# Religious Freedom and Education in Australian Schools

Paul Babie 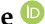

Adelaide Law School, The University of Adelaide, Adelaide, SA 5005, Australia; paul.babie@adelaide.edu.au

**Abstract:** This article examines the constitutional allocation of power over primary and secondary education in Australia, and the place of and protection for freedom of religion or belief (FoRB) in Australian government and religious non-government schools. This article provides both an overview of the judicial treatment of the constitutional, legislative, and common law protection for FoRB and a consideration of emerging issues in religious freedom in both government and religious non-government schools, suggesting that the courts may soon be required to provide guidance as to how the available protections operate in both settings.

**Keywords:** Australia; education; freedom of religion; constitution; free exercise; section 116; implied freedom of political communication

## 1. Introduction

This article examines the place of and protection for freedom of religion or belief (FoRB) in Australian government primary and secondary education ('government schools') and in religious non-government schools.[1] In order to understand that place, it is necessary to outline the legal foundations for education in Australia as well as the protection of FoRB in Australian law. Australia, like the United States, is a constitutional federal democracy; as such, the legal foundations for education involve an uneasy balance of federal (national, or Commonwealth) and state and territory (regional) law and policy. This means that no one unitary source exists for the foundation of government schools, its funding, and for the protection of FoRB. Instead, both government and religious non-government schools derive protection of FoRB through a piecemeal and tenuous collection of constitutional, legislative, and common law provisions, at both the Commonwealth and state and territory levels.

This chapter contains five parts. Part 2 outlines the legal status of government primary and secondary education and in religious non-government schools in the Australian states and territories. The purpose of Part 3 is two-fold: first, it provides an overview of constitutional, legislative, and common law protection for FoRB within the multi-cultural, multi-faith environment found in Australia, and, second, it examines the judicial application of those protections in the government and religious non-government school settings; in fact, as will be seen, there is very little judicial analysis. As such, it is necessary to extrapolate from the general approach taken by the courts to the available FoRB protections. Part 4 examines emerging issues in religious and in government schools, suggesting that the courts may soon be required to provide guidance as to how the available protections operate in both the religious and government school settings. Part 5 offers brief concluding reflections.[2]

## 2. Legal Foundation for Education

Australian students generally begin school with a kindergarten or preparatory school year, and then undertake 12 years of primary and secondary schooling; ten years of

---

1   In Australia, 'public education' is referred to as 'government schools'.

2   Some of the material in this article appears earlier and in other forms, for which see (Babie 2021).

---

schooling is generally the minimum, although there is some variation between states and territories as to school starting and leaving ages. A fundamental distinction exists in Australian law between government and non-government[3] primary,[4] secondary,[5] and tertiary[6] education. State legislation provides the framework within which each operates.[7] Notwithstanding the state and territory foundation for government schools, however, significant funding for both government and non-government, including religious, schools comes from the states and territories and from the Commonwealth.[8]

Commonwealth legislation provides for primary and secondary school funding on the basis of the Schooling Resource Standard (SRS), 'a measure of the amount of public funding needed by each school to meet the educational needs of its students.'[9] Once the requirements of the SRS are met, funding depends upon the state or territory entering a bilateral agreement with the Commonwealth for minimum state and territory funding requirements. Moreover, s 22A of the *Australian Education Act 2013* (Cth) requires that states and territories must meet or exceed minimum funding contribution requirements for both government and non-government sectors as a condition of receiving Commonwealth funding.[10] State and territory governments have discretion to fund above these requirements.[11]

The legal foundation for religious education differs based upon the type of school. Thus, secular non-government schools usually eschew specifically religious education in favour of broader cultural, social and historical studies subjects. The majority of religious non-government schools appear to advocate a values-based approach to religious education, with a greater or lesser focus on evangelism or proselytism depending upon the individual school. Perhaps the greatest single difference between religious and government schools is the religious backdrop to instruction, and the corresponding intensity of explicitly religious activities (chapels, devotions, prayers, or similar) in non-teaching time during the day.[12]

Special Religious Instruction (SRI), while available in all Australian government schools, receives variable treatment depending on the state or territory.[13] South Australia provides a broadly representative example of Australian state government approaches to SRI.[14] Provision for religious education in South Australian Government schools is made by the *Education Act 1972* (SA), with the content and conduct of religious education in Government schools provided for by the *Education Regulations 2012* (SA) made under the Act.[15] Among other things, the *Regulations* provide for the establishment of standing committees[16] and school-specific Religious Education Committees,[17] and for the devel-

---

3　Variously referred to as 'independent', 'religious', 'church' or 'private', depending on the particular school.

4　'Primary' schools correspond approximately to US elementary schools, typically catering for students from Reception to Grade 6 or 7.

5　Secondary schools typically cater for students from Grade 7 or 8 to Grade 12 or 13.

6　Tertiary educaiton is that which occurs in institutions of technical and further education (TAFE) and universities or higher education providers.

7　The South Australian legislation, which is representative of the state legislation, is found in the *Education Act 1972* (SA) and *Education Regulations 2012* (SA).

8　The Commonwealth legislation, which applies to all states is the *Australian Education Act 2013* (Cth); *Australian Education Regulation 2013* (Cth). The South Australian legislation is the *Education Act 1972* (SA) and *Education Regulations 2012* (SA). See also (Australian Government Department of Education, Skills and Employment 2013, 2021).

9　(Australian Government Department of Education, Skills and Employment 2013).

10　Ibid.

11　Ibid.

12　(Rowe 2017).

13　See, e.g., *Education and Training Reform Act 2006* (Vic) s 2.2.11. See also (Halafoff 2013; Barker 2014; Halafoff and Bouma 2019).

14　Relevant legislation in the states and territories includes *Education Act 2004* (ACT); *Education Act 1990* (NSW); *Education Act 2015* (NT); *Education (General Provisions) Act 2006* (Qld); *Education Act 2016* (Tas); *Education and Training Reform Act 2006* (Vic); *School Education Act 1999* (WA).

15　See (Department of Education and Children's Services and South Australian Government 2013).

16　*Education Regulations 2012* (SA) Pt 7.

17　Ibid.

opment of course modules for distribution to schools.[18] Most significantly, government schools also provide for the conduct of both 'religious education' and 'religious seminars' in schools,[19] which are typically 'opt-out/opt-in' at the discretion of parents, or conducted outside of school hours. If students opt-out, in some cases, ethics courses may be offered as substitutes for SRI.[20]

### 3. Protection for Freedom of Religion or Belief (FoRB)

One finds what protections exist for FoRB in the Australian government school context scattered across Commonwealth and state constitutional texts, Commonwealth and state and territory legislation, and the common law. This part considers each in turn, including its judicial treatment. Significantly, very limited judicial treatment of these protections in the educational context exists; as such, the application to that context explored here is largely based upon extrapolation from the existing judicial authority, which is itself sparse.

*3.1. Commonwealth and State Constitutions*

Australian constitutional texts establish two types of protection for FoRB in the government school environment: express protection found in the text itself, and freedoms implied from that text, of which only one has yet been developed by the courts, that for political communication.

3.1.1. Express Protection

Section 116 of the *Australian Constitution* provides four FoRB guarantees:

The Commonwealth shall not make any law for establishing any religion, or for imposing any religious observance, or for prohibiting the free exercise of any religion, and no religious test shall be required as a qualification for any office or public trust under the Commonwealth.

Section 46 of the *Constitution Act 1934* of the state of Tasmania provides a similar protection:

(1)　Freedom of conscience and the free profession and practice of religion are, subject to public order and morality, guaranteed to every citizen.

(2)　No person shall be subject to any disability, or be required to take any oath on account of his religion or religious belief and no religious test shall be imposed in respect of the appointment to or holding of any public office.

The High Court has interpreted s 116 as a limitation on the legislative and executive power of the Commonwealth, as opposed to a broad protection of an individual right.[21] Taking a similar approach, the Tasmanian Supreme Court calls s 46 a 'historical puzzle'[22] which 'does not, in terms, confer any personal rights or freedoms on citizens.'[23] As such, both s 116 and s 46 seem to provide, at best, no more than a qualified guarantee for FoRB.

The High Court has, though, taken a broad approach to what might be protected, as religion, within the ambit of s 116. In *Adelaide Company of Jehovah's Witnesses Incorporated v Commonwealth*,[24] Latham CJ, wrote that:

---

18　Ibid.

19　Ibid.

20　(Rowe 2017).

21　*Attorney-General (Vic); Ex rel Black v Commonwealth* ('*DOGS Case*') (1981) 146 CLR 559, 605. A vast scholarly literature has developed around the interpretation of s 116: see, e.g., (Aroney et al. 2017; Babie 2007, 2011, 2015, 2017, 2018, 2020; Babie and Bhanu 2018; Babie and Rochow 2010; Babie et al. 2019; Babie and Krumrey-Quinn 2014; Barker 2015a, 2015b, 2020b, 2020d; Beck 2016a, Beck 2016b; Beck 2013; Krieg and Babie 2013; Langos and Babie 2020).

22　(Tasmanian Constitutional Law Reform Project 2021).

23　*Corneloup v Launceston City Council* [2016] FCA 974, [38].

24　*Adelaide Company of Jehovah's Witnesses Incorporated v Commonwealth* (1943) 67 CLR 116.

> It would be difficult, if not impossible, to devise a definition of religion which would satisfy the adherents of all the many and various religions which exist, or have existed, in the world. There are those who regard religion as consisting principally in a system of beliefs or statement of doctrine. So viewed religion may be either true or false. Others are more inclined to regard religion as prescribing a code of conduct. So viewed a religion may be good or bad. There are others who pay greater attention to religion as involving some prescribed form of ritual or religious observance. Many religious conflicts have been concerned with matters of ritual and observance. Section 116 must be regarded as operating in relation to all these aspects of religion, irrespective of varying opinions in the community as to the truth of particular religious doctrines, as to the goodness of conduct prescribed by a particular religion, or as to the propriety of any particular religious observance. What is religion to one is superstition to another. Some religions are regarded as morally evil by adherents of other creeds. At all times there are many who agree with the reflective comment of the Roman poet—"*Tantum religio potuit suadere malorum.*" The prohibition in s. 116 operates not only to protect the freedom of religion, but also to protect the right of a man to have no religion. No Federal law can impose any religious observance. Defaults in the performance of religious duties are not to be corrected by Federal law—Deorum injuriae Diis curae. Section 116 proclaims not only the principle of toleration of all religions, but also the principle of toleration of absence of religion.[25]

Thus, both religion, and the right to be free from it, are broadly protected by s 116. This may prove important, as we will see, when we turn to emerging issues in Part IV.

Yet, it is difficult to say what protections s 116 and s 46 may create in the government school educational context; s 46 has never been applied to that setting, and only the establishment and religious test guarantees of s 116 have been applied to government schools.[26] In *Attorney-General (Vic); Ex rel Black v Commonwealth (Defence of Government Schools)* (*DOGS case*) the High Court held that Commonwealth funding of non-government schools does not constitute 'establishing [a] religion' for the purposes of s116.[27] In *Williams v Commonwealth* (*School Chaplains Case*)[28] the Court held that a religiously affiliated chaplain funded by the Commonwealth to provide a government school chaplaincy program did not violate the religious test for a Commonwealth office guarantee of s 116. The Commonwealth's funding was 'insufficient to render a chaplain engaged . . . the holder of an office under the Commonwealth'.[29]

Taken together, the *DOGS* and *School Chaplains* cases constitutionalise interaction and cooperation between the state and both government and religious non-government schools, providing for a "semi-permeable membrane" or "imaginary wall" rather than an impenetrable barrier between church and state: 'metaphorically, the flow of Commonwealth largesse to religious institutions is permitted; what is blocked is the reverse passage of religious entanglement with Commonwealth affairs'.[30]

The most significant guarantee found in s 116, however, that for free exercise, has never been applied in the educational setting. Indeed, since federation in 1901, only three High Court decisions have dealt with the free exercise guarantee,[31] confirming the same test enunciated in *Krygger v Williams* ('*Krygger*'):

---

[25] Ibid. 123. See also (Puls 1998); cf. (Deagon and Saunders 2020).

[26] Although Renae Barker's important work provides some guidance as to the application of the religious freedom protections in the school setting: see (Barker 2020a, 2020c).

[27] *DOGS Case* (1981) 146 CLR 559.

[28] *Williams v Commonwealth* ('*School Chaplains Case*') (2012) 248 CLR 156.

[29] Ibid. [110].

[30] (Frame 2006; MacFarlane and Fisher 1971), citing *Lemon v. Kurtzman*, 403 U.S. 602, 613 (1971).

[31] *Krygger v Williams* (1912) 15 CLR 366; *Adelaide Company of Jehovah's Witnesses Inc v Commonwealth* (1943) 67 CLR 116; *Kruger v Commonwealth* (1997) 190 CLR 1 ('*Kruger*').



> To require a man to do a thing which has nothing at all to do with religion is not prohibiting him from a free exercise of religion. It may be that a law requiring a man to do an act which his religion forbids would be objectionable on moral grounds, but it does not come within the prohibition ... [32]

The judicial use of this test means that laws regulating matters that have 'nothing at all to do with religion',[33] and laws that have as their effect the infringement of religion but do not 'discriminate against religion generally'[34] do not violate this prohibition. It therefore seems, at a minimum, that in the government school setting, s 116 protects against at least direct infringements of FoRB, as well as attempts to establish a defined state religion, to which observance must be given, and of which one must be a member in order to hold a public office.

Of course, an individual right to FoRB must be understood as operating within the wider community. For that reason, limitations may justifiably be placed upon the former to protect the interests of the latter. In *Adelaide Co of Jehovah's Witnesses Inc v The Commonwealth* ('*Jehovah's Witnesses*') Latham CJ said that:

> the protection of any form of liberty as a social right within a society necessarily involves the continued existence of that society as a society. Otherwise the protection of liberty would be meaningless and ineffective. It is consistent with the maintenance of religious liberty for the State to restrain actions and courses of conduct which are inconsistent with the maintenance of civil government or prejudicial to the continued existence of the community. The Constitution protects religion within a community organized under a Constitution, so that the continuance of such protection necessarily assumes the continuance of the community so organized. This view makes it possible to reconcile religious freedom with ordered government. It does not mean that the mere fact that the Commonwealth Parliament passes a law in the belief that it will promote the peace, order and good government of Australia precludes any consideration by a court of the question whether or not such a law infringes religious freedom. The final determination of that question by Parliament would remove all reality from the constitutional guarantee. That guarantee is intended to limit the sphere of action of the legislature. The interpretation and application of the guarantee cannot, under our Constitution, be left to Parliament. If the guarantee is to have any real significance it must be left to the courts of justice to determine its meaning and to give effect to it by declaring the invalidity of laws which infringe it and by declining to enforce them. The courts will therefore have the responsibility of determining whether a particular law can fairly be regarded as a law to protect the existence of the community, or whether, on the other hand, it is a law 'for prohibiting the free exercise of any religion.'[35]

But this approach to free exercise has never been applied in the school setting, and so it remains unclear the extent to which it may protect religious education in the government school setting. Still, given the level of funding for SRI in state and territory government schools, there does seem the potential for increased concern about the protection of FoRB. And it does seem clear that the Commonwealth may justify both the limitation of some rights inherent in FoRB as well as the funding for the teaching of SRI as justifiable and consistent with the provision of ordered government and the protection of the existence of the community. Indeed, it may be that this pragmatic judicial approach to s 116 allows the Commonwealth to conclude a functioning outcome with the states and territories

---

[32]   *Krygger v Williams* (1912) 15 CLR 366, 369.

[33]   Ibid.

[34]   *Church of the New Faith v Commissioner of Pay-Roll Tax (Vic)* (1983) 154 CLR 120.

[35]   *Adelaide Co of Jehovah's Witnesses Inc v The Commonwealth* (1943) 67 CLR 116, 131–2.

concerning education, balancing the community interest against the individual right to FoRB.[36]

### 3.1.2. Implied Protection

In addition to the express guarantees found in the text of the Commonwealth (and perhaps the Tasmanian) *Constitution*, the High Court has also found implied in the text of the latter a protection for freedom of political communication.[37] Implications from the text which support a freedom of political communication may provide additional scope for the protection of FoRB in the educational context. In *McCloy v New South Wales*,[38] the High Court stated the implied freedom of political communication as a:

[a] qualified limitation on legislative power implied in order to ensure that the people of the Commonwealth may 'exercise a free and informed choice as electors.' It is not an absolute freedom. It may be subject to legislative restrictions serving a legitimate purpose compatible with the system of representative government for which the Constitution provides, where the extent of the burden can be justified as suitable, necessary and adequate, having regard to the purpose of those restrictions.[39]

What is significant in relation to FoRB is that this freedom of political communication is the result of implications drawn by the judiciary from the *text* of the *Constitution* itself.

Justice Lionel Murphy was the first to give expression to the fact that '[t]he Constitution is a framework for a free society'[40] and that, as such, rights could be implied from that framework. Justice Murphy, over thirty years ago, wrote this about the framework established by the Australian *Constitution*:

Traditionally, constitutions are instruments which briefly state the framework of government, the political divisions and organs, their composition, functions and interrelations, and sometimes specific guarantees of human rights. Because of the brevity of constitutions, implications are a prominent feature in the history of their judicial interpretation. The Australian Constitution does not express all that is intended by it: much of the greatest importance is implied. Some implications arise from consideration of the text; others arise from the nature of the society which operates the constitution. Constitutions are designed to enable a society to endure through successive generations and changing circumstances . . .

A constitutional principle, such as responsible government, may even appear inconsistent with the written text, nevertheless it operates . . . Even where specific rights are spelled out, for example, in the United States Constitution, there may remain others which are implied . . .

. . .

The history of interpretation of the Australian Constitution shows that implications have been freely made. Implications of federalism, in particular of intergovernmental immunity, have been made, but these are not the only possible implications.

. . .

---

36 See (Grenfell and Moulds 2018).

37 *Nationwide News Pty Ltd. v Wills* (1992) 177 CLR 1; *Australian Capital Television v Commonwealth* (1992) 177 CLR 106; *Theophanous v Herald & Weekly Times Ltd.* (1994) 182 CLR 104; *Stephens v West Australian Newspapers Ltd.* (1994) 182 CLR 211; *Lange v Australian Broadcasting Corporation* (1997) 189 CLR 520; *APLA Ltd. v Legal Services Commissioner* (NSW) (2005) 224 CLR 322; *Hogan v Hinch* (2011) 243 CLR 506; *Unions NSW v State of New South Wales* (2013) 88 ALJR 227; *Attorney-General (SA) v Corporation of the City of Adelaide* (2013) 249 CLR 1; *McCloy v New South Wales* [2015] HCA 34; *Brown v Tasmania* (2017) 261 CLR 328; *Clubb v Edwards; Preston v Avery* [2019] HCA 11.

38 *McCloy v New South Wales* [2015] HCA 34, [2].

39 Ibid. (footnotes and citations omitted).

40 *Seamen's Union of Australia v Utah Dev Co* (1978) 144 CLR 120, 158.

> In my opinion, other constitutional implications which are at least as important as that of responsible government, arise from the nature of Australian society. The society professes to be a democratic society—a union of free people, joined in one Commonwealth with subsidiary political divisions of States and Territories. From the nature of our society, an implication arises prohibiting slavery or serfdom. Also from the nature of our society, reinforced by the text . . . in my opinion, an implication arises that the rule of law is to operate, at least in the administration of justice. Again, from the nature of our society, reinforced by parts of the written text, an implication arises that there is to be freedom of movement and freedom of communication. Freedom of movement and freedom of communication are indispensable to any free society. . . . The implication raised is not of absolute freedom, but it is at least freedom from arbitrary interference.[41]

Based upon this democratic principle, Justice Murphy thus found fundamental rights implied in the *Constitution*. He elaborated upon one of those rights, the freedom of political communication, in *Ansett Transport Industries (Operations) Pty. Ltd. V. Commonwealth*:

> Elections of federal Parliament provided for in the Constitution require freedom of movement, speech and other communication, not only between the States, but in and between every part of the Commonwealth. The proper operation of the system of representative government requires the same freedoms between elections. These are also necessary for the proper operation of the Constitutions of the States . . . From the [express] provisions and from the concept of the Commonwealth arises an implication of a constitutional guarantee of such freedoms, freedoms so elementary that it was not necessary to mention them in the Constitution . . . The freedoms are not absolute, but nearly so. They are subject to necessary regulation (for example, freedom of movement is subject to regulation for purposes of quarantine and criminal justice; freedom of electronic media is subject to regulation to the extent made necessary by physical limits upon the number of stations which can operate simultaneously). The freedoms may not be restricted by the Parliament or State Parliaments except for such compelling reasons.[42]

The nexus between text and implied right is therefore important. And it matters especially in the case of FoRB for the fact that of the few express rights found in the *text* of the *Constitution*, FoRB is one of them, in s 116. It is at least arguable that the implication of a freedom of political communication must also include, to some extent, the freedom to communicate about religious matters, provided that the speech touches upon 'choices to be made by the people of the Commonwealth as the sovereign political authority.'[43] Indeed, the High Court itself supported this conclusion in *Attorney-General (SA) v Corporation of the City of Adelaide*,[44] in which it held that:

> some 'religious' speech may also be characterised as 'political' communication for the purposes of the freedom . . . Plainly enough, preaching, canvassing, haranguing and the distribution of literature are all activities which may be undertaken in order to communicate to members of the public matters which may be directly or indirectly relevant to politics or government at the Commonwealth level. The class of communication protected by the implied freedom in practical terms is wide.[45]

It would appear, therefore, that the implication of the freedom of political communication can itself draw a further implication from the fact of an express protection for FoRB in s 116

---

41　*McGraw-Hinds (Austl) Pty Ltd. v Smith* (1979) 144 CLR 633, 668–70.

42　*Ansett Transp Indus (Operations) Pty Ltd. v Commonwealth* (1977) 139 CLR 54, 88 (citations omitted).

43　*Clubb v Edwards; Preston v Avery* [2019] HCA 11, 29. And see also (Landrigan 2014).

44　*Attorney-General (SA) v Corporation of the City of Adelaide* (2013) 249 CLR 1, 43–4, 73–4.

45　Ibid.

such that religious speech can be political speech for the purposes of the implied freedom. Thus, the ambit of the implied freedom of political communication can expand to cover religious communication.[46]

More importantly, and unlike the express rights, which apply only as against the Commonwealth, because the implied freedoms are drawn from the Constitution as a whole, which, as Justice Murphy makes clear, establish a democratic principle applicable to the Commonwealth and to the states, the freedom of political communication may limit not only to the Commonwealth, but also the states.[47] The ambit of the implied freedom of political communication may be taken, then, to cover religious speech, whether the infringement is Commonwealth or state. Of course, some restrictions upon expression are always necessary for the purposes of maintaining the community interest to be protected against harmful speech. The High Court has judicially crafted a three-step standard by which to test the justifiability of limitations which may be placed upon the implied freedom by either the Commonwealth or the states (see also Figure 1. Assessing Infringements of the Implied Freedom of Political Communication Pursuant to *McCloy v New South Wales*):

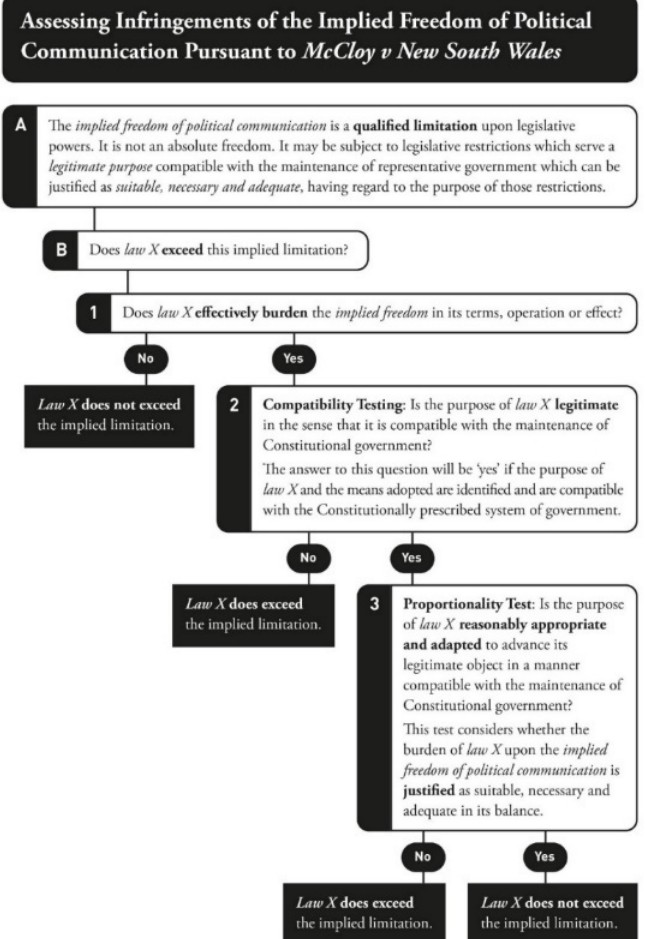

**Figure 1.** Assessing Infringements of the Implied Freedom of Political Communication Pursuant to McCloy v New South Wales[48].

---

[46] On the potential for judicial creativity which may allow for the use of implications to be drawn from s 116, see (Detmold 1994); cf. (Stone 2005). And see (Langos and Babie 2020).

[47] This is now well established: *Unions NSW v State of New South Wales* (2013) 252 CLR 530; *Attorney General (SA) v Corp of the City of Adelaide* (2013) 249 CLR 1; *McCloy v New South Wales* (2015) 257 CLR 178; *Brown v Tasmania* (2017) 261 CLR 328; *Clubb v Edwards; Preston v Avery* [2019] HCA 11; *Spence v. Queensland* [2019] HCA 15.

[48] © Nigel Williams 2021.

the question whether an impugned law infringes the freedom requires application of the following propositions derived from previous decisions of this Court and particularly *Lange v Australian Broadcasting Corporation*[49] and *Coleman v Power*:[50]

The freedom under the Australian Constitution is a qualified limitation on legislative power implied in order to ensure that the people of the Commonwealth may "exercise a free and informed choice as electors." It is not an absolute freedom. It may be subject to legislative restrictions serving a legitimate purpose compatible with the system of representative government for which the Constitution provides, where the extent of the burden can be justified as suitable, necessary and adequate, having regard to the purpose of those restrictions.

The question whether a law exceeds the implied limitation depends upon the answers to the following questions, reflecting those propounded in *Lange* as modified in *Coleman v Power*:

Does the law effectively burden the freedom in its terms, operation or effect?

If "no", then the law does not exceed the implied limitation and the enquiry as to validity ends.

If "yes" to question 1, is the purpose of the law legitimate, in the sense that it is compatible with the maintenance of the constitutionally prescribed system of representative and responsible government?[51] This question reflects what is referred to in these reasons as "compatibility testing".

The answer to that question will be in the affirmative if the purpose of the law and the means adopted are identified and are compatible with the constitutionally prescribed system in the sense that they do not adversely impinge upon the functioning of the system of representative government.

If the answer to question 2 is "no", then the law exceeds the implied limitation and the enquiry as to validity ends.

If "yes" to question 2, is the law reasonably appropriate and adapted to advance that legitimate object in a manner that is compatible with the maintenance of the constitutionally prescribed system of representative and responsible government?[52] This question involves what is referred to . . . as "proportionality testing" to determine whether the restriction which the provision imposes on the freedom is justified.

The proportionality test involves consideration of the extent of the burden effected by the impugned provision on the freedom. There are three stages to the test—these are the enquiries as to whether the law is justified as suitable, necessary and adequate in its balance in the following senses:

*Suitable*—as having a rational connection to the purpose of the provision;

*Necessary*—in the sense that there is no obvious and compelling alternative, reasonably practicable means of achieving the same purpose which has a less restrictive effect on the freedom;

*Adequate in its balance*—a criterion requiring a value judgment, consistently with the limits of the judicial function, describing the balance between the importance of the purpose served by the restrictive measure and the extent of the restriction it imposes on the freedom.

---

49　*Lange v Australian Broadcasting Corporation* (1997) 189 CLR 520.

50　*Coleman v Power* (2004) 220 CLR 1.

51　This version of the question was substituted by *Brown v Tasmania* (2017) 261 CLR 328, [104].

52　Ibid.

If the measure does not meet these criteria of proportionality testing, then the answer to question 3 will be "no" and the measure will exceed the implied limitation on legislative power.[53]

Neither the implied freedom, nor the standard for assessing limitations which may be placed upon it has been tested in the educational setting. It may be assumed, though, that the freedom, if it does extend to religious belief and speech in the form of worship, may both protect religious speech within the government school setting, and prevent proselytising as a justifiable limitation imposed upon such speech.

### 3.2. Commonwealth and State Legislation

Two forms of legislative protection exist for FoRB: bills or charters of rights enacted by the states of Victoria and Queensland, and by the Australian Capital Territory, and Commonwealth, state and territory anti-discrimination legislation. While neither form of protection has been expressly applied in the educational setting, this section considers the judicial application of each to FoRB generally.

### 3.2.1. Bills or Charters of Rights

The *Human Rights Act 2004* (ACT), *The Charter of Human Rights and Responsibilities Act 2006* (Vic), and the *Human Rights Act 2019* (Qld) protect a range of fundamental human rights,[54] including express protections for freedom of thought, conscience, religion and belief, and the rights of minorities to enjoy their own culture, religion and language.[55] In addition to these specific religious rights, the legislation also protects the rights to equality before the law,[56] life,[57] privacy,[58] peaceful assembly and association,[59] expression, taking part in public life,[60] and liberty and security of the person.[61] These rights apply only to individuals,[62] and 'may be subject to reasonable limits set by . . . laws that can be demonstrably justified in a free and democratic society.'[63]

Surprisingly, citizens have made little use of the rights protected by these legislative bills or charters of rights[64] and, as such there is almost no judicial analysis of any of the rights protected, and none at all of the FoRB protections. It remains to be seen, then, how protections might apply to FoRB generally, and in the educational setting specifically. It seems likely, though, that should they be used, the protections would provide the same sort of protections for those who seek to exercise religious freedom or to be free from proselytising in government schools as are found in s 116 of the *Constitution*.

### 3.2.2. Anti-Discrimination or Equality

Commonwealth and state and territory human rights and anti-discrimination legislation provide negative protections and positive obligations in respect of equality. Commonwealth equality protections—known as 'federal discrimination law'—protect every Australian against unlawful discrimination[65] pursuant to four principal enactments: *Racial*

---

53   *McCloy v New South Wales* (2015) 257 CLR 178, [2] (citations omitted). And see also *Brown v Tasmania* (2017) 261 CLR 328.

54   See (Byrnes et al. 2008); (Evans and Evans 2008); (Groves and Campbell 2017).

55   See, e.g., *Human Rights Act 2004* (ACT) ss 14 and 27.

56   Ibid. s 8.

57   Ibid. s 9.

58   Ibid. s 12.

59   Ibid. s 15.

60   Ibid. s 16.

61   Ibid. s 18.

62   Ibid. s 6.

63   Ibid. s 28(1).

64   See (Aroney et al. 2017).

65   (Australian Human Rights Commission 2016).



*Discrimination Act 1975* (Cth), *Sex Discrimination Act 1984* (Cth), *Disability Discrimination Act 1992* (Cth), and the *Age Discrimination Act 2004* (Cth).[66] While a proposed *Religious Discrimination Bill 2019* (Cth) is currently before the Commonwealth Parliament,[67] federal discrimination law currently provides no positive protection against discrimination on religious grounds.

The states and territories,[68] however, provide comprehensive protection against defined grounds of discrimination, including FoRB.[69] This legislation both prohibits discrimination against individuals on the basis of a professed religious belief[70] or the lack of such a belief[71] and exempts religious orders, bodies or institutions generally (including religious and non-religious educational institutions) allowing for discrimination on prohibited grounds necessary for the religious purposes of the order, body, or institution.[72]

While the exemptions take on paramount importance in non-government religious schools in relation to the hiring of teachers,[73] in government schools the equality and non-discrimination principle provides protection for religious expression and for the freedom from proselytising for those who hold no religion. Yet these protections have been raised in only one government school case. *Aitken v Victoria* involved a challenge to the use of SRI in Victorian government schools, was ultimately unsuccessful, and provides no guidance as to the application of non-discrimination in the government school setting.[74] The litigation did, however, prompt a change in Victorian government policy, making SRI opt-in/opt-out during after-school hours.[75]

At best, we can assume that the rights protecting discrimination on religious grounds may prevent attempts at discrimination on religious grounds in religious non-government schools, and against proselytising in the government school environment. It is likely, though, that the most significant outcome of challenges pursuant to anti-discrimination legislation will be to pressure governments to alter its policies concerning SRI, as has been the case in Victoria.[76]

### 3.3. Common Law

While the Australian common law contains no express recognition of rights,[77] canons of statutory interpretation provide limited protection through a 'common law bill of rights'

---

66    To this list could be added a fifth, the *Fair Work Act 2009* (Cth).

67    See (Australian Government, Attorney-General's Department 2019).

68    *Anti-Discrimination Act 1977* (NSW); *Equal Opportunity Act 1984* (SA); *Equal Opportunity Act 1984* (WA); *Discrimination Act 1991* (ACT); *Anti-Discrimination Act 1991* (Qld); *Anti-Discrimination Act 1996* (NT); *Anti-Discrimination Act 1998* (Tas); *Equal Opportunity Act 2010* (Vic).

69    A list of the prohibited grounds, representative of state and territory regimes, found in the *Equal Opportunity Act 2010* (Vic) s 6, includes: age; breastfeeding; employment activity; gender identity; disability; industrial activity; lawful sexual activity; marital status; parental status or status as a carer; physical features; political belief or activity; pregnancy; race; religious belief or activity; sex; sexual orientation; an expunged homosexual conviction; and, personal association (whether as a relative or otherwise) with a person who is identified by reference to any of the these attributes.

70    See, e.g., *Equal Opportunity Act 2010* (Vic) s 6(1)(n). In Victoria, discrimination on the basis of a characteristic of a person's religion is also prohibited: *Equal Opportunity Act 2010* (Vic) s 7(2)(b) and (c); *Kapoor v Monash University* (2001) 4 VR 483.

71    See, e.g., *Equal Opportunity Act 2010* (Vic) s 4(1).

72    See, e.g., *Anti-Discrimination Act* 1977 (NSW) ss 31A, 31K, 46A, 49ZO, and 56; *Sex Discrimination Act 1984* (Cth) ss 37, 38; *Equal Opportunity Act 1984* (SA) ss 50, 85ZM; *Equal Opportunity Act 1984* (WA) ss 66(1), 72, 73; *Discrimination Act 1991* (ACT) ss 32, 33, 46; *Anti-Discrimination Act 1991* (Qld) ss 41(a), 90, 109; *Anti-Discrimination Act 1996* (NT) ss 30(2), 37A, 51; *Anti-Discrimination Act 1998* (Tas) ss 51, 52; *Equal Opportunity Act 2010* (Vic) ss 39, 81, 82, 83, 84. See also *Fair Work Act 2009* (Cth) ss 153(2)(c), 195(2)(c), 351 (2)(c), 772(2)(c); *Sex Discrimination Act 1984* (Cth) s 37(a), (b), (d), and 38.

73    For the judicial treatment of the exemptions as concern FoRB, see *OV & OW* [2010] NSWCA 155; *Christian Youth Camps Ltd. v Cobaw Community Health Service Ltd.* [2014] VSCA 75.

74    In a challenge to SRI in Victorian government schools, the Victorian Civil and Administrative Tribunal (VCAT) found no discrimination: see *Aitken v State of Victoria (Department of Education & Early Childhood Development (Anti-Discrimination)* [2012] VCAT 1547. An appeal against this decision was dismissed by the Supreme Court of Victoria Court of Appeal: *Aitken v State of Victoria* [2013] VSCA 28 (22 February 2013). Another challenge was settled before it was decided by the VCAT: see (Sehee 2019). For the background to some of the challenges which have been mounted, but not judicially resolved, see (Humanists Victoria 2009).

75    (Sehee 2019).

76    (Humanists Victoria 2009).

77    *Grace Bible Church v Reedman* (1984) 36 SASR 376.

and the principle of legality.[78] These norms apply to legislation which may purport to limit a specific right, such as FoRB. In assessing a statute which seeks to limit FoRB, a court considers whether it does so in clear and unambiguous terms.[79] In assessing ambiguous legislation, the court favours a construction which is most in conformity with Australia's treaty obligations concerning the right in question.[80] As with constitutional and legislative protection for FoRB, there exists no judicial analysis of the application of these canons to the government school setting. It may be assumed, then, that should the courts be called upon to analyse legislation pursuant to these canons, they will do so in accordance with these general principles.

## 4. Emerging Issues

It may seem that FoRB enjoys robust constitutional, legislative, and common law protection in the school setting. Upon closer inspection, however, it is clear that a number of questions remain unanswered, in both the religious and government school settings. This part considers three emerging or potential issues: public health emergencies, such as COVID-19, online speech, and freedom *from* religion.

The difficulty, of course, remains that the courts continue to show limited interest in making broad use of the available FoRB protections against legislative and executive encroachments. Moreover, there is no judicial analysis at all of the operation of these principles in the educational context, with the exception of the *School Chaplains Case* and *Aitken v Victoria*, both of which provide no guidance as to the application of s 116 or anti-discrimination legislation to the government school setting. There is little doubt, though, that the courts will soon be called upon to analyse the way in which the constitutional and legislative protections apply in the government school setting.

### 4.1. Public Health Emergencies (COVID-19)

Throughout 2020, governments the world over responded to the COVID-19 pandemic. In the United States, in order to address the public health dimension of the crisis, a number of states limited attendance at worship services, while simultaneously allowing activities to continue in bars, casinos, and even strip clubs.[81] In the latter part of 2020, California, Kentucky, and Michigan closed schools. In *Monclova Christian Academy v. Toledo-Lucas County Health Department*,[82] the United States Court of Appeals for the Sixth Circuit dealt with a case from Ohio involving a county health department order closing all schools, including faith-based schools, so as to slow the spread of COVID-19. Gyms, tanning salons, office buildings, and a large casino were permitted to remain open. Nine Christian schools successfully challenged the closure on the basis that it constituted a violation of the free exercise clause of the First Amendment to the US Constitution. Is it possible that a similar challenge could arise in Australia and, if so, how would the courts deal with it?

---

[78]   (Spiegelman 2008).

[79]   *Church of the New Faith v Commissioner for Pay-roll Tax (Vic)* (1983) 154 CLR 120, 130; *Re Bolton; Ex Parte Douglas Beane* (1987) 162 CLR 514, 523; *Canterbury Municipal Council v Moslem Alawy Society Ltd.* (1985) 1 NSWLR 525, 544.

[80]   *Minister for Immigration and Ethnics Affairs v Teoh* (1995) 183 CLR 273, 287.

[81]   See (Russo 2021; Babie and Russo 2020a, 2020b).

[82]   *Monclova Christian Academy v Toledo-Lucas County Health Department*, 2020 WL 7778170 (6th Cir, Dec. 31, 2021). See also (Blackman 2021). The Supreme Court of the United States also rejected a challenge from religious schools that objected to pandemic-related state orders in Kentucky: *Danville Christian Academy, Inc., et al v. Andy Beshear, Governor of Kentucky*, 592 U. S. ____ (2020).

Australia followed the global trend concerning Covid-19, with the Commonwealth, states, and territories all imposing sweeping closures at various times during 2020.[83] Taking South Australia as a representative example, the *Emergency Management Act 2004* (SA), s 25, empowers the Police Commissioner to issue emergency orders to address public health emergencies, including the closure of both government and religious non-government schools, as has indeed occurred in a number of cases.[84] It is unclear whether a challenge to such a closure would succeed in the Australian courts, given the current interpretation of s 116.

As we have seen, using the current analysis of s 116 in *Krygger*, *Jehovah's Witnesses*, and *Kruger*, an infringement of free exercise depends upon whether the closure order had the express purpose of limiting free exercise. Thus, in the educational setting, an order would require as its direct purpose the closure of religious schools. Orders issued in 2020 did not have as their express purpose such a limitation of free exercise. Rather, those orders were general in nature, and applied to religious organizations, including non-government religious schools, in the same way that they did to secular institutions and businesses. A challenge to such an order would undoubtedly fail under the current interpretation. If, however, as I have argued elsewhere, a wider approach to s 116 was taken, in which the effect of the order was taken as capable of infringing the protection, it is possible that such a challenge could succeed, at least in the sense that a violation of free exercise would be found.[85] But if that was merely the first step of a two-step process found in Latham CJ's judgment, then a court would be required to determine if the limitation, for public health reasons, was justifiable. It seems likely that such limitations could be upheld, if for public health reasons and if the challenged limitation applied equally to all institutions, both religious and secular.

### 4.2. Online Speech

In *B.L. v Mahanoy Area School Dist.*,[86] the United States Court of Appeal for the Third Circuit found that student speech outside of school premises, hours, and publications is fully protected under the First Amendment unless it falls within a recognised exception, such as speech involving threats.[87] The Third Circuit wrote that:

> the primary responsibility for teaching civility rests with parents and other members of the community. As arms of the state, public schools have an interest in teaching civility by example, persuasion, and encouragement, but they may not leverage the coercive power with which they have been entrusted to do so. Otherwise, we give school administrators the power to quash student expression deemed crude or offensive—which far too easily metastasizes into the power to censor valuable speech and legitimate criticism. Instead, by enforcing the Constitution's limits and upholding free speech rights, we teach a deeper and more enduring version of respect for civility and the "hazardous freedom" that is our national treasure and "the basis of our national strength."[88]

---

83  For a comprehensive and current list of the relevant emergency orders imposed in every Australian jurisdiction, see (JusticeConnect 2021). The South Australian emergency order is representative: *Emergency Management Act 2004* (SA), s 25, authorises the Police Commissioner to issue emergency orders, pursuant to which the following directions have been issued: *Emergency Management (Public Activities No 16) (COVID-19) Direction 2020*, 14 December 2020; *Emergency Management (Cross Border Travel No 22) (COVID-19) Direction 2020*, 14 December 2020; *Emergency Management (COVID-19) (Isolation Following Diagnosis or Close Contact) Direction 2020*, 8 September 2020; *Emergency Management (COVID-19) (Peppers Waymouth Isolation) Direction 2020*, 17 November 2020; *Emergency Management (CODI-19) (Parafield Cluster Isolation No 4) Direction 2020*, 28 November 2020; *Emergency Management (Supervised Quarantine) (COVID-19) Direction 2020*, 14 December 2020; *Emergency Management (Residential Aged Care Facilities No 17) (COVID-19) Direction 2020*, 14 December 2020; *Emergency Management (Prohibition of Point of Care Serology Tests) (COVID-19) Direction 2020*, 24 June 2020; *Emergency Management (Reporting on COVID-19 Testing No 2) Direction 2020*, 9 September 2020.

84  See (Government of South Australia 2020).

85  (Babie 2017). And see also (Klonick 2018; Smith 2021).

86  *B.L. v Mahanoy Area School Dist.*, No 19-1842 (MD Pa No 3:17-cv-01734) (30 June 2020).

87  (Volokh 2020).

88  *B.L. v Mahanoy Area School Dist.*, No 19-1842 (MD Pa No 3:17-cv-01734) (30 June 2020), 44 (footnotes omitted).

This case involved, as the court put it, a snapchat that was 'crude, rude, and juvenile'; it had nothing to do with religion. What I want to suggest here, though, and drawing on this case for the Australian setting, is simply this: would student religious speech, outside of school premises, hours, and publications, and which the school then seeks to limit, be protected by the implied freedom of political communication? Whether the school is a government or a non-government school, the fact of Commonwealth funding for both means that an attempt to limit such non-school-setting and school-time speech could, depending upon the nature of the speech, offend the implied freedom of political communication.

As we know, the implied freedom of political communication may cover religious speech. It is likely that the implied freedom covers speech made in the online environment.[89] Australian states and territories have taken a range of approaches to the use of digital technology in schools, from acceptable use policies to outright bans during school hours and on school premises.[90] The issue in Australia would turn on whether such speech could be characterized as 'political' for the purposes of the implied freedom.

A threshold issue, however, may arise due to the age of the person whose speech is restricted. While it may seem plausible that political communication is possible only amongst those old enough to vote (and thus permitted to take part in the democratic process), this requirement is not one which the High Court has, as yet, placed upon the implied freedom. And, in any case, to so constrain the freedom would seem unnecessarily to limit its scope in a way that fails to comport with the reality of democracy and politics. Rather, it is at the very least arguable that many classes of citizens can engage in political speech without holding the legal right to vote. Consider two examples. First, women prior to gaining the franchise in many Western democracies. Surely the substance of the speech associated with the suffragette movement would have constituted political speech capable of protection under the implied freedom had it been available at the time. Indeed, such speech has a very long lineage. In Aristophanes's *Lysistrata*, a play from Greek antiquity, the women of warring cities decide to withhold sexual privileges from their husbands and lovers in an ultimately successful effort to force them to negotiate peace.[91] Surely the plan to withhold those privileges constitutes a form of speech capable of protection under a freedom of political communication. Or, to take a second example, and without in any way equating this with women seeking the franchise, what of persons convicted of criminal offences while serving jail sentences. Some states limit or revoke the right to vote for such persons. Does that class of persons thereby also lose the freedom of political communication in protesting against the loss of the franchise? Surely not. While students may be treated as falling into a category of persons who otherwise could vote, but are denied that right due to gender or to having lost the status of a person entitled to vote, there is nothing to prevent a polity from lowering the age of majority, such that the right may be extended to students. That fact alone suggests that the freedom of political communication must protect all citizens, and not merely those denied the right to vote due either to circumstances over which they have no control—age or gender—or those over which they do have control—criminal conviction—but which may change over time as social conditions change. To deny the freedom on the basis of such circumstances seems at best misguided and at worst discriminatory in a way that the implied freedom itself seems designed to overcome.

Thus, assuming that a person is entitled to the right, and that whatever speech could be characterized as political speech, then it is certainly possible that any limitation of such speech by a school, whether government or non-government, may run afoul of step one of the *McCloy* test. But would a student be able to demonstrate that the speech was not reasonably limited under the second and third parts of the *McCloy* test? No guidance as

---

89  See (Langos and Babie 2020); (Australian Human Rights Commission 2016). Although it raised the issue of the implied freedom operating in the online environment, the High Court refused to answer the question in *Smethurst v Commissioner of Police* [2020] HCA 14.

90  See (de Zwart 2018; Shannon 2020; AAP 2019).

91  (Aristophanes 1990).

to how this inquiry might proceed exists, and so it remains an open question. Given the trend of political speech cases in Australia, however, the prospects of success of a student challenge seem limited.

*4.3. Freedom from Religion*

Fairness in Religions in School (FIRIS), which operates in New South Wales and Victoria, 'is a group of parents and citizens who want education about religions in state schools that is':[92]

> Delivered by teachers employed by education departments without bias, using materials in the Australian Curriculum
> Respectful of the secular nature of state schools
> Consistent with Australia's multicultural society
> Inclusive, not divisive or discriminatory
> Committed to fostering citizenship.

FIRIS supports the NSW Department of Education's statement that 'schools are neutral grounds for rational discourse and objective study', and that public schools should not be arenas for 'opposing political views or ideologies'. The Department allows religious organizations to do things other organizations cannot.[93]

Similarly, the Humanist Society of Victoria (HSV) seeks 'to ensure that religious privilege/overreach/discrimination does not infringe upon the rights of people who do not identify as religious to be able to access, as well as to provide, spiritual care.' The HSV has been involved in litigation in Victoria, and supports similar action in New South Wales, the object of which is to require that government school chaplains may be of any or of no faith.[94]

The issue raised by FIRIS and HSV is whether there is a protection in Australian law to be free *from* religion. As we have seen, Latham CJ countenanced the possibility that 'the prohibition in s. 116 operates not only to protect the freedom of religion, but also to protect the right of a man to have no religion.'[95] What little scholarship that is on this point remains divided: some take the view that Latham CJ's position was correct, and that s 116 protects the right both to hold a religion and to hold no religion;[96] others argue that s 116, in both its history and its purpose, protects only the right to hold a religion.[97] As the number of such challenges increases, we await judicial clarification.

**5. Conclusions**

Australian law provides for the establishment of both government and non-government religious schools. Two paradoxes, however, arise. On the one hand, while one might assume that the latter schools are secular, in many cases they are not, with many of them providing some form of either religious of values-based education. The High Court has given constitutional approval to these arrangements. Similarly, in the case of non-government religious schools, one might assume that these schools would be separate, at least as a matter of funding, from both Commonwealth and state governments. In many cases, though, they are not. And, thus, again, with approval from the High Court, this arrangement has been given constitutional imprimatur. And as a result, FoRB issues arise in both types of school.

In determining the extent of FoRB protection in the Australian educational setting, however, two issues create difficulties. First, there is very limited judicial interpretation

---

92 (Fairness in Religions in School 2019).

93 Ibid.

94 (Sehee 2019).

95 *Adelaide Company of Jehovah's Witnesses Incorporated v Commonwealth* (1943) 67 CLR 116, 123.

96 (Puls 1998).

97 (Deagon and Saunders 2020).

of the various possible protections: the express constitutional protection of s 116 and its implied corollary, the freedom of political communication, the anti-discrimination legislation of the Commonwealth and state and territory governments, and the common law. Compounding this problem for educators is the fact that of this limited judicial interpretation of the possible protections for FoRB, almost none of it deals with the educational setting. Thus, for the most part, the protection for FoRB for students is largely a matter of speculation.

Still, it is possible to foresee novel issues on the horizon with which the courts may be forced to grapple. This article canvasses three: the possibility for public health restrictions to limit FoRB, for school policies to limit online student religious speech, either in school or outside of school premises and hours, and for religious instruction in government schools to violate the right to be free from religion. Given the paucity of judicial authority, however, as with the protection of FoRB itself, while we may speculate, the outcome of such challenges remains to be seen.

**Funding:** This research received no external funding.

**Institutional Review Board Statement:** Not applicable.

**Informed Consent Statement:** Not applicable.

**Data Availability Statement:** Not applicable.

**Conflicts of Interest:** The author declares no conflict of interest.

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
