# Peer review of "Religious Freedom and Education in Australian Schools"

_laws, 2020_

Round 1
Reviewer 1 Report
See attached.

Author Response
I am very grateful to the Reviewer for these very helpful comments. The Reviewer's suggestions are copied bellow, with my changes, which have been made in track-changes, and the page number in the revised manuscript noted below, in bold type.
Reviewer's Comments:
1. I am wondering if there could be a chart or some type of figure to help the reader follow the questions related to: Does the law effectively burden the freedom in its terms, operation or effect? -- I defer to the author, but a figure here would present the nuance of this judicial test.
I have included a figure for this purpose: see pages 9 and 10.
2. When discussing Monclova Christian Academy v. Toledo-Lucas County Health Department, you might note that the school did prevail in that case. citing to the actual case. It might also be noted that the U.S. Supreme Court rejected a challenge from religious schools that objected to pandemic-related state orders in Kentucky (see https://www.nytimes.com/2020/12/17/us/politics/supreme-court-religiousschools-coronavirus.html).
These changes have been made: see pages 14 and note 78.
3. The author used footnotes throughout, and I think there might need to be in-text citations in Chicago Style. I will defer to the editor.
I will defer to the Editors as to whether this change needs to be made.
Typo: In the United States, in order to address the public health dimension of the crisis, a number states limited attendance at worship services, while simultaneously allowing activities to continue in bars, casinos, and even strip clubs.
This change has been made: see page 14.
Reviewer 2 Report
The paper examines Freedom of Religion and Belief (ForB) in the Australian context for government and non-government (predominantly religious schools). This is an important and complex topic. The debate about religion in schools was at a peek prior to the outbreak of COVID-19 and, while paused as a result of the pandemic, has not been forgotten. However, the paper falls into a common trap of trying to do too much and as a result, the impact of the arguments and research is lost.
The paper accurately highlights the lack of judicial examination of ForB in the Australian educational setting and thus by necessity attempts to extrapolate from existing judicial decision in other areas and academic opinion. However, given the paucity of directly relevant judicial opinions the paper needs to engage more with existing literature that examines the relationship between ForB and education in Australia. While the author cites some of the leading commentators, others are missing. For example, Dr Luke Beck’s, and Professors Paul Babie and Nicholas Aroney are all not referenced, amongst others. While Dr Renae Barker’s work is cited her work specifically on freedom of religion in education is absent. While I fully acknowledge that it is not possible to cite every relevant published paper on a topic the absence of judicial opinions on this topic necessitates a greater engagement with the literature.
While the paper is on FoRB the largest section examines the constitutionally implied right to freedom of political communication. While this right IS relevant to freedom of religion, and the author explains this, the paper feels un-balanced. Should the author wish to revise the paper I would recommend confining it to examining the role that the freedom of political communication could play in protecting FoRB in Australian schools. Thus confined the author could examine this right in much more depth and provide a more original contribution to the scholarship. Overviews of FoRB in schools in Australian already exist but the role of the freedom of political communication is an area that has not had significant scholarly attention. Amongst the issues that would need to be considered is whether those under the age of voting (as the vast majority of school children are) can engage in political communication.
Finally, there are two specific examples used in the paper that felt odd. First, the paper refers to primary and tertiary education. A footnote indicates that this is intended to refer to children from predatory school through to grades 12 or 13. However, in Australian this is usually referred to as primary and secondary education with the term tertiary education reserved for University education. Second the paper occasionally gives examples from South Australia. This may well be the author’s home jurisdiction. However, such examples need to be prefaced by “for example” or similar or some explanation provided that demonstrate that South Australia is representative of the approach in other jurisdictions.
While I see merit in this paper I am of the view that it needs to be further refined and tightened before publication.
Author Response
I am very grateful to the Reviewer for these very helpful comments. The Reviewer's suggestions are copied bellow, with my changes, which have been made in track-changes, and the page number in the revised manuscript noted below, in bold type.
Reviewer's Comments:
The paper accurately highlights the lack of judicial examination of ForB in the Australian educational setting and thus by necessity attempts to extrapolate from existing judicial decision in other areas and academic opinion. However, given the paucity of directly relevant judicial opinions the paper needs to engage more with existing literature that examines the relationship between ForB and education in Australia. While the author cites some of the leading commentators, others are missing. For example, Dr Luke Beck’s, and Professors Paul Babie and Nicholas Aroney are all not referenced, amongst others. While Dr Renae Barker’s work is cited her work specifically on freedom of religion in education is absent. While I fully acknowledge that it is not possible to cite every relevant published paper on a topic the absence of judicial opinions on this topic necessitates a greater engagement with the literature.
I have added footnotes to account for this literature: see pages 4-5, notes 19 and 24.
While the paper is on FoRB the largest section examines the constitutionally implied right to freedom of political communication. While this right IS relevant to freedom of religion, and the author explains this, the paper feels un-balanced. Should the author wish to revise the paper I would recommend confining it to examining the role that the freedom of political communication could play in protecting FoRB in Australian schools. Thus confined the author could examine this right in much more depth and provide a more original contribution to the scholarship. Overviews of FoRB in schools in Australian already exist but the role of the freedom of political communication is an area that has not had significant scholarly attention. Amongst the issues that would need to be considered is whether those under the age of voting (as the vast majority of school children are) can engage in political communication.
With the greatest of respect to the Reviewer, and on advice from the Editor assigned to this issue, I will decline this invitation. I feel that what matters in my article is an overview of religious freedom as it is protected in Australian law, and as it applies to the educational context. Given the piecemeal nature of the former, and the paucity of the latter, I feel that the article makes a significant contribution on that basis. To accept this invitation from the Reviewer would be to write an entirely different paper, which is not my project. I have, however, adopted the Reviewer's very important point that it may be necessary for the courts to consider the age of a student in considering the issue of political communication. I do, though, feel that to limit the freedom only to those who are entitled to vote would unnecessarily limit the scope of the implied freedom in a way that the High Court has not required. See page 17.
Finally, there are two specific examples used in the paper that felt odd. First, the paper refers to primary and tertiary education. A footnote indicates that this is intended to refer to children from predatory school through to grades 12 or 13. However, in Australian this is usually referred to as primary and secondary education with the term tertiary education reserved for University education. Second the paper occasionally gives examples from South Australia. This may well be the author’s home jurisdiction. However, such examples need to be prefaced by “for example” or similar or some explanation provided that demonstrate that South Australia is representative of the approach in other jurisdictions.
I have made change with respect to the issue of primary, secondary, and tertiary education: see page 2, notes 3-5
With respect to the second issue, the use of South Australia as an example, in both cases where I used that example, the necessary language was already in the text: see page 3, lines 73-5, and page 16, line 516.
Round 2
Reviewer 2 Report
The changes made to the paper add to an already excellent paper on this topic. The addition of additional references adds a lot to the paper and will make it an even stronger resource and point of reference for those wishing to learn about or conduct further research on this topic.
While I still have some reservations about the balance of the paper this is a matter of style and I note as per the author's comments that this has been discussed with the editors.
I note the author has added references to “secondary education” to the paper as recommended in my previous report. There are a few placed where the paper still refers to primary and tertiary education only (for example line15 and 27 – 28). I suggest a final proof read of the paper to catch any further instances to help avoid confusion on the part of future readers of the paper.